# AutoMixer: A Lightweight and Scalable Industrial 5.0 Safety Assurance Model with Multi-Scale Adaptive Dual-Attention

## Abstract

With the rapid growth of intelligent transportation and industrial automation, traffic safety management and industrial system safety generate vast amounts of spatio-temporal data. These data offer rich temporal and spatial patterns for analysis but pose significant challenges, including dynamic traffic patterns, high-dimensional sensor data, and complex anomalies in industrial systems. Traditional methods struggle to capture nonlinear accident patterns, handle noisy sensor data, or model intricate multi-variable interactions, especially in real-time scenarios. Although deep learning and large-scale models have improved the accuracy of accident prediction and anomaly detection, their reliance on complex spatial operations and large parameter sizes creates computational bottlenecks, limiting scalability in large-scale and real-time safety applications. Therefore, we propose AutoMixer, a lightweight and scalable model that avoids explicit spatial modeling. It uses a dual cross-attention module to identify coupled trend and periodic features in multi-resolution spatio-temporal data. Extensive experiments demonstrate that AutoMixer consistently outperforms state-of-the-art baselines, achieving 7% higher detection accuracy while effectively handling large-scale node distributions and high-frequency data. AutoMixer provides a practical and deployable solution for real-time accident detection and industrial system safety analysis, enhancing computational efficiency and applicability in resource-constrained environments, thus optimizing performance for large-scale traffic and industrial safety tasks.

## 1 Introduction

With the rapid advancement of intelligent transportation systems and industrial automation, environments from highways to production lines produce vast amounts of diverse spatiotemporal data. Effective analysis of historical data enables the detection of traffic accidents, hazardous road conditions and industrial system anomalies like equipment failures or process deviations, crucial for enhancing traffic safety and industrial reliability. Consequently, large-scale operational data analysis is central to traffic safety management and industrial system safety. However, the complexity of spatiotemporal data presents significant challenges, including high-frequency data streams, dynamic accident patterns, data heterogeneity and noise in industrial sensor logs. Traditional deep learning methods struggle to capture dynamic accident characteristics or multi-scale anomalies, particularly when spatial dependencies from road topology or sensor layouts are weak or unclear Ali et al. (2025).

For instance, traffic accident data often lacks clear topological correlations due to complex road conditions or mixed traffic flows. Similarly, industrial sensor data typically lacks inherent spatial structure. Furthermore, limited computational resources and discontinuous spatiotemporal data streams heighten the challenge of detecting accidents or anomalies in large-scale road networks or sensor arrays. This makes long-term prediction and anomaly detection from short input sequences a critical need. Therefore, developing efficient long-term spatiotemporal prediction methods for anomaly detection without explicit spatial modeling is a research priority. Such models must enhance detection accuracy and meet the stringent real-time analysis demands of large-scale traffic safety and human-machine collaborative Industry 5.0 safety applications, thereby supporting proactive accident prevention, system reliability, and data-driven decision-making for urban system evaluation.

Existing spatiotemporal analysis methods often enforce explicit spatial modeling, introducing unnecessary computational overhead. These approaches fail to significantly improve detection accuracy for accidents or anomalies while diminishing overall computational efficiency. Models like DCRNN Li et al. (2017), STHGCN Wang et al. (2022), and TCGCN Wang et al. (2024a), which rely on graph convolutional networks to capture spatial dependencies, exhibit model complexities of at least $O(n)$, resulting in substantially higher computational costs for high-dimensional datasets. However, they offer limited improvements in modeling dynamic safety patterns. Similarly, methods such as StSVGP Hamelijnck et al. (2021) and StGLMM Anderson et al. (2022) face a trade-off between expressiveness and efficiency in large-scale traffic or industrial safety benchmarks. The use of spatial kernel functions in these methods often escalates computational complexity to $O(N^3)$, leading to memory overflows or degraded detection accuracy in real-time applications.

The large scale of spatial nodes in traffic safety and industrial system safety datasets, e.g., traffic accident records, vehicle sensor data, and industrial equipment logs, forms spatiotemporal networks with tens of thousands of nodes. In intelligent traffic safety systems, IoT sensors at intersections collect high-frequency data at second intervals Sotres et al. (2017), with cities often exceeding ten thousand nodes. Similarly, industrial systems, such as the Electricity Transformer Temperature or Tennessee Eastman datasets, generate high-volume sensor streams. Spatiotemporal prediction models with spatial dependency modules face sharply increased computational complexity, which limits scalability and parallelizability for real-time accident or anomaly detection. In large-scale forecasting tasks for traffic safety and industrial systems, high-frequency data, extensive node distributions, limited resources, and discontinuous data require models to handle short input sequences (about 100 steps) and predict over horizons of up to 400 steps.

These constraints demand both high predictive accuracy and computational efficiency. This study introduces AutoMixer, a deep learning model that eliminates explicit spatial modules to balance prediction accuracy, scalability across numerous nodes, computational efficiency, and real-time applicability in Industry 5.0 contexts. The AutoMixer model consists of four core components: adaptive frequency-domain decomposition, dynamic coupled feature weighting, a dual-attention mechanism for spatiotemporal analysis, and a multi-resolution dynamic coupling module. Empirical results demonstrate that AutoMixer achieves accurate spatiotemporal predictions without relying on explicit spatial information decomposition, outperforming state-of-the-art baselines. It shows superior efficiency, robustness, and practicality in real-world Industry 5.0 scenarios. The key contributions of AutoMixer include enhanced efficiency and scalability for real-time accident and anomaly detection in traffic safety and Industry 5.0 system reliability applications, summarized as follows:

- **Adaptive Frequency-Domain Decomposition of spatio-temporal Sequences**: AutoMixer utilizes the Discrete Fourier Transform (DFT) to adaptively decompose the frequency-domain representations of spatio-temporal sequences across multiple resolutions. This enables effective extraction of key frequency components and cross-scale temporal patterns. Compared to traditional techniques such as moving averages, this approach offers superior scale-awareness in complex datasets, while naturally supporting channel independence and facilitating multi-task generalization.

- **Multi-Resolution Dynamic Coupling Weights**: This module assigns trainable, dynamic weights to features across multiple temporal resolutions, enabling adaptive representation of multi-scale spatio-temporal patterns. By optimizing interactions between different resolutions, it enhances the model's ability to generate accurate long-term forecasts from short input sequences. The design is particularly well-suited for real-time, large-scale node analysis, offering both flexibility and efficiency.

- **Dual-Attention for Modeling Spatio-Temporal Sequences**: This module integrates dual attention mechanisms to capture cross-attention between periodic and trend components across multiple resolution scales. A dynamic feature decoding module is embedded at the output stage to further enhance temporal representation. This design significantly improves the model's ability to make accurate predictions under weak spatial dependencies, making it particularly effective for industrial production and traffic safety scenarios.

- AutoMixer achieves superior performance in long-horizon prediction from short input sequences across diverse spatio-temporal datasets. It consistently reduces prediction error by approximately 1.2%–4.7% compared to all baselines, while maintaining computational efficiency suitable for real-time applications. Its strong predictive power, especially in weakly

spatial domains such as transportation and energy systems, highlights its robustness, scalability, and practical deployment potential.

## 2 METHODOLOGY

### 2.1 PROBLEM DEFINITION

We formulate anomaly and security risk detection for data lacking explicit spatial information as a multivariate time series prediction problem, focusing on capturing complex temporal dependencies efficiently. Significant deviations between actual and predicted data indicate anomalies, with larger deviations implying higher anomaly probabilities. Given a historical sequence of length $L$, denoted as $\{x_1, x_2, \ldots, x_L\}$, where each $x_t \in \mathbb{R}^M$ is an $M$-dimensional observation vector at time $t$, the goal is to predict the next $T$ time steps: $\{x_{L+1}, x_{L+2}, \ldots, x_{L+T}\}$, with each $x_{L+t} \in \mathbb{R}^M$.

For datasets with spatial information, the task extends to multivariate spatiotemporal forecasting, using a sequence $\{X_1, X_2, \ldots, X_L\}$, where each $X_t \in \mathbb{R}^{N \times M}$ captures $M$ variables across $N$ spatial nodes (e.g., sensors). An adjacency matrix $A \in \mathbb{R}^{N \times N}$ encodes spatial relations like connectivity. The objective is to forecast the next $T$ steps $\{X_{L+1}, X_{L+2}, \ldots, X_{L+T}\}$, with each $X_{L+t} \in \mathbb{R}^{N \times M}$, leveraging historical data, spatial structure $A$, and latent spatiotemporal interactions.

### 2.2 PROPOSED FRAMEWORK

AutoMixer is a novel model tailored for time series forecasting, specifically designed to handle short input sequences while enabling accurate long-range predictions. The overall architecture consists of three main components: *Input Initialization Module*, *AutoMixer Attention Block* and *Output Predictive Module*, which together form a lightweight yet powerful forecasting framework. As illustrated in Fig. 1, AutoMixer processes the input sequence through adaptive embedding and frequency-domain decomposition, followed by a human-machine information hybrid attention mechanism and resolution-aware coupling strategy required by Industry 5.0 and generates long-term predictions.

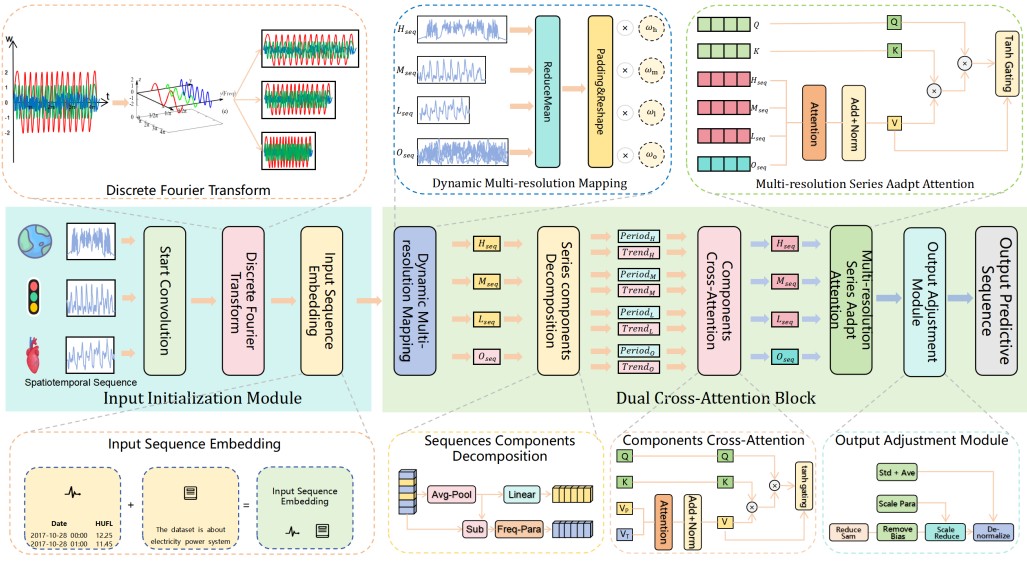

Figure 1: The Framework of AutoMixer

### 2.2.1 INPUT INITIALIZATION MODULE

The forecasting process starts with an input initialization module using a Conv1D layer to compress multivariate time series into compact representations for hybrid forecasting. The sequence then undergoes DFT-based frequency-domain decomposition, converting it into multiscale sub-series with

distinct frequency components. By retaining the top-$k$ components from high-, mid-, and low-frequency bands, noise is filtered out, preserving key signal structures, enhancing hybrid analysis quality, and improving prediction accuracy and computational efficiency in high-frequency and large-scale node environments, which is defined as:

$$X(k) = \sum_{n=0}^{N-1} x(n)\, e^{-j\frac{2\pi}{N}nk}, \quad k = 0, 1, \ldots, N-1, \tag{1}$$

$$x(n) = \frac{1}{N} \sum_{k=0}^{N-1} X(k)\, e^{j\frac{2\pi}{N}nk}, \quad n = 0, 1, \ldots, N-1, \tag{2}$$

where $x(n)$ denotes the $n$-th time-domain sample, $X(k)$ represents the $k$-th frequency-domain coefficient, $N$ is the sequence length, and $j$ is the imaginary unit. Compared to traditional time-domain techniques such as moving averages, this approach more effectively aligns with the periodicity and oscillatory characteristics commonly observed in real-world spatio-temporal data.

The AutoMixer input sequence embedding module transforms raw multivariate time series data of shape $X \in \mathbb{R}^{B \times T \times C}$ into embedded representations of shape $X \in \mathbb{R}^{B \times T \times d_{\text{model}}}$, enabling efficient downstream decomposition and attention-based modeling. It incorporates positional encoding to preserve temporal structure and integrates dataset-specific metadata, such as data length and configuration-derived characteristics, for dynamic adaptation to diverse datasets. This design promotes channel independence and task-aware feature embedding, enhancing modeling efficiency and accuracy in weak or non-spatial scenarios, thus improving robustness and adaptability across real-world forecasting tasks.

$$X_{\text{R-Emb}} = \text{Linear}(X_{\text{input}}) + \text{PosEnc}(X_{\text{input}}), \tag{3}$$

$$\text{PosEnc}(pos, 2i) = \sin\left(\frac{pos}{10000^{2i/d_{\text{model}}}}\right), \tag{4}$$

$$X_{\text{Emb}} = X_{\text{R-Emb}} + \text{Task}(\text{pred\_len}). \tag{5}$$

The input sequence embedding module serves as a bridge between raw multivariate time series data and the subsequent spatio-temporal attention analysis. It effectively enhances the model's ability to capture spatio-temporal dependencies. Empirical results indicate that setting the embedding dimension $d_{\text{model}} = 32$ achieves an optimal balance between representational capacity and computational efficiency, particularly suitable for tasks involving short input sequences and long-term predictions. However, the success of this embedding relies heavily on the quality of positional encoding to preserve temporal order and avoid information loss.

### 2.2.2 Dynamic Multi-resolution Projection

The data flows through this module following initialization. It generates, aligns, and dynamically weights multi-scale sequences starting from the raw input $X \in \mathbb{R}^{B \times T \times C}$. The time dimension is compressed to extract detailed features for each resolution scale using adaptive average pooling, which preserves key characteristics while minimizing information loss.

Next, the module pads the multi-scale sequences along the time dimension to ensure uniform length across all scales. Specifically, for each sequence $X_m$ with length $T_m < T_{\max} = \max_m\{\text{len}(X_m)\}$, the last time step value is repeated instead of padding with zeros, avoiding noise or interference. This maintains continuity in periodic components and ensures consistent input dimensions for subsequent attention mechanisms. Formally, the padding operation is defined as:

$$\text{pad} = X_m[:,-1:,:] \cdot \varepsilon \otimes \text{repeat}(1, T_{\max} - T_m, 1), \tag{6}$$

$$X'_m = \text{cat}([X_m, \text{pad}], \dim = 1), \tag{7}$$

where $T_{\max} - T_m$ indicates the padding length, and the small scaling factor $\varepsilon$ prevents abrupt discontinuities in the sequence.

The model employs a **Dynamic Scale Attention** module to generate dynamic scale weights $\omega_{\text{scale}-\text{m}}$, enabling adaptive weighting of multi-scale data feature contributions. The process begins by flattening the input sequence $X \in \mathbb{R}^{B \times T \times C}$ into a one-dimensional vector, followed by feature

extraction through a multi-layer perceptron (MLP). Temporal dependencies are then captured using an attention mechanism to compute the dynamic scale weights. The process is formalized as:

$$X_{\text{scale}-\text{m}} = \sigma\left(W_2\,\sigma(W_1\,\text{flatten}(X) + b_1) + b_2\right), \tag{8}$$

$$\omega_{\text{scale}-\text{m}} = \text{Softmax}\left(\text{MLP}\left(\text{Softmax}\left(\frac{QK^\top}{\sqrt{d_k}}\right)V\right)\right), \tag{9}$$

where $\sigma$ denotes the ReLU activation function; $W_1, W_2$ and $b_1, b_2$ are trainable weights and biases ensuring the feature dimension matches $d_{\text{model}}$; $Q$, $K$, and $V$ are the query, key, and value matrices projected from multi-head features (4 heads), effectively capturing temporal correlations.

The **Sequence Components Decomposition** module in AutoMixer decomposes multi-scale spatio-temporal sequences $X \in \mathbb{R}^{B \times T \times C}$ into *trend* and *periodic* components, enabling independent extraction and fusion of multi-scale spatio-temporal features. This decomposition is repeatedly applied within the Dual Cross-Attention Block, controlled by learnable parameters, and supports channel-wise independence across different scales. The **trend component** is extracted via average pooling, followed by a non-linear activation and a linear transformation:

$$X_m^T = \text{Linear}\left(\sigma\left(\text{AvgPool}(X_m)\right)\right), \tag{10}$$

where $\sigma(\cdot)$ is the activation function, and $X_m$ denotes the input at scale $m$.

The **periodic component** is computed by subtracting the trend from the original input and multiplying element-wise with a learnable frequency modulation parameter $\mathbf{F}$:

$$X_m^P = \sigma\left(X_m - \sigma\left(\text{AvgPool}(X_m)\right)\right) \otimes \mathbf{F}, \tag{11}$$

where $\otimes$ denotes element-wise multiplication. This decomposition allows effective separation of trend and periodic signals, enhancing spatio-temporal representation learning.

### 2.2.3 Adaptive Dual Cross-Attention Mechanism

The core component of AutoMixer is the Adaptive Dual Cross-Attention Mechanism, which comprises two cross-attention operations combined with an adaptive decoding analysis module. The first cross-attention operates on spatio-temporal sequences decomposed into trend and periodic components, coupling features across four channels in a scale-aware manner: trend features are processed from low to high resolution, while periodic features are handled from high to low resolution. This enables effective multi-scale feature interaction. The mathematical formulation is given by:

$$X_m' = \text{Norm}\left(Q + \left(\text{softmax}\left(\frac{QK^\top}{\sqrt{d_k}}\right)V\right) \times \tanh(\gamma)\right), \tag{12}$$

where $Q_p$, $K$, and $V = V_t \times V_p$ denote the query, key, and value matrices, respectively; $d_k$ is the dimension of the key vectors; the softmax function normalizes attention scores along each row; $\gamma$ is a learnable parameter that stabilizes the attention scaling during early training; and *Norm* refers to the layer normalization operation.

Subsequently, the **Adaptive Decoding Analysis Module** integrates multi-scale predictive features. It takes concatenated predictive sequence features along with padding as input, which are then processed by a performance evaluator to produce performance scores, and by an uncertainty evaluator to estimate deviation variances $\mathbf{U}$. These outputs serve as dynamic parameters within the adaptive decoding module, enabling flexible decoding of multi-scale spatio-temporal sequence features for improved prediction accuracy. The corresponding formulations are as follows:

$$\text{Scores}(X_m') = \mathbf{W}_2\left(\text{Dropout}\left(\mathbf{W}_1\sigma(X_m' + \mathbf{b}_1), p\right)\right) + \mathbf{b}_2, \tag{13}$$

$$\mathbf{U}(x) = \ln\left(1 + \exp\left(\mathbf{W} \cdot x + \mathbf{b}\right)\right), \tag{14}$$

$$X_m^* = X_m' \times \text{Softmax}\left(\text{Scores}(X_m') \times \exp\left(-\mathbf{U}(X_m')\right)\right), \tag{15}$$

where $x_m'$ denotes the input feature vector with time dimension $T_{\text{total}}$; $\mathbf{W}_1$ and $\mathbf{b}_1$ are the weight matrix and bias vector of the first linear transformation, mapping from $T_{\text{total}}$ to $T_{\text{hidden}}$; Dropout randomly zeroes elements with probability $p$; and $\mathbf{W}_2$ and $\mathbf{b}_2$ correspond to the second linear transformation from $T_{\text{hidden}}$ to $T_{d_{\text{model}}}$.

This module dynamically integrates predictive feature scores and uncertainty variances from multi-scale spatio-temporal components during the decoding process, significantly enhancing prediction accuracy and stability in complex spatio-temporal sequence forecasting.

Finally, the second cross-attention operation couples four groups of multi-resolution spatio-temporal features, mathematically formulated as:

$$Y' = \text{Norm}\left( Q + \left( \text{softmax}\left( \frac{QK^\top}{\sqrt{d_k}} \right) V \right) \times \tanh(\gamma) \right), \tag{16}$$

$$V = V_H \times V_M \times V_L \times V_O, \tag{17}$$

where $V_H$, $V_M$, $V_L$, and $V_O$ represent the value matrices corresponding to high, medium, low, and other resolution features, respectively.

### 2.2.4 Output Adjustment Module

The output adjustment module refines the model's multi-scale prediction sequences through a sequence of normalization reversal steps. First, the predictions are debiased and proportionally scaled to match the dimensions of the initial convolutional layer (Start Conv). Then, they are denormalized by multiplying by the standard deviation and adding the mean, restoring the outputs to their original scale. This procedure ensures that final predictions align accurately with target dimensions, maintaining both accuracy and stability across various anomaly detection tasks.

## 3 Experiments

We employ a diverse set of datasets, including industrial production safety data sequences such as Ett, Electricity, and Weather, as well as traffic safety detection datasets like Metr-LA, PEMS-BAY, Traffic, and PEMS03/08. These datasets cover a wide range of Industry 5.0 safety detection scenarios, providing a robust benchmark for evaluating the performance and scalability of the proposed large-scale models. The experimental settings, including comparison methods, verification metrics, environment, and parameters, are detailed in the appendix.

### 3.1 Experimental Results

#### 3.1.1 Baseline Comparison experiments

**Industrial Production Safety Detection Datas Comparison Experiment.** We evaluate AutoMixer on the long-term sequences anomaly prediction task using the industrial production safety detection datasets. The input length is 96 and the prediction horizons are set to 96, 192, 336, and 720 steps. The Appendix A.2.4 Table 5 shows that AutoMixer outperforms existing methods, including PatchTST, Transformer-based models, and traditional approaches. It achieves strong results across electricity, solar energy, and weather forecasting tasks. Compared to the original TimeMixer, AutoMixer shows clear improvements due to its adaptive frequency-domain decomposition and multi-resolution coupling design, making it effective for long-sequence prediction.

**Traffic Operation Safety Detection Datas Comparison Experiment.** We evaluate AutoMixer on a long-term spatio-temporal anomaly-risk detection task using traffic operation safety detection datasets. As shown in Appendix A.2.5 Table 6, AutoMixer consistently achieves lower MSE and MAE across different horizons, showing strong predictive performance. With its dual-attention dynamic analysis framework, AutoMixer effectively captures spatio-temporal patterns, leading to robust and accurate forecasts.

#### 3.1.2 Ablation Study

We conducted ablation studies to evaluate AutoMixer's key components: frequency-domain adaptive decomposition, multi-scale dynamic coupling, dual attention, and dynamic prediction ensemble. As shown in Fig. 2, the dual-attention module captures cross-scale dependencies, boosting performance. Omitting the prediction ensemble lowers robustness and long-term accuracy. Adaptive decomposition separates trend and periodic components, while fixed coupling weights reduce adaptability. These findings underscore each design's role in enhancing anomaly detection and prediction accuracy.

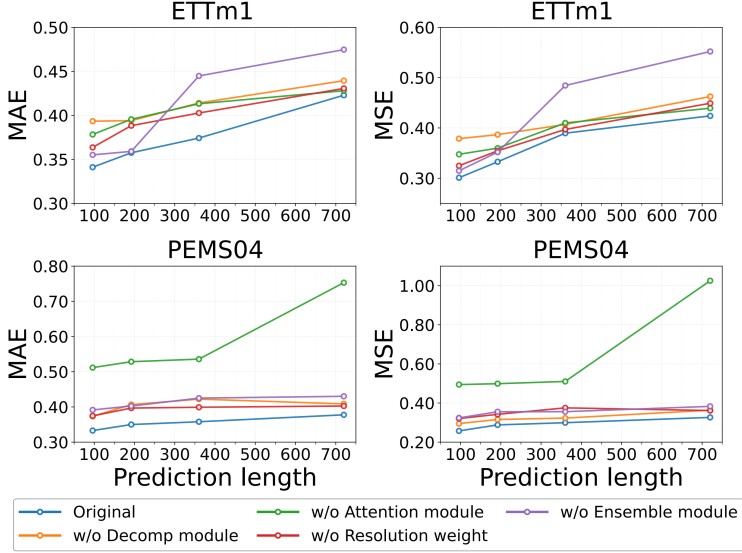

Figure 2: Results Of Ablation Experiments Under Multiple Prediction Steps

### 3.1.3 COMPONENTS STUDY

Table 1 presents the evaluation of five key components in AutoMixer. The spatio-temporal decomposition effectively separates trend and periodic components, with the adaptive average module yielding the best results. Feature extraction via component decomposition and integration performs best with a 3-layer structure, while deeper models risk overfitting. Among decomposition methods, DFT achieves the highest accuracy. The attention mechanism uses a cross-attention strategy—prioritizing periodic components from fine to coarse and trends from coarse to fine—to enhance fusion. Finally, integrating all resolution data with an adaptive strategy leads to optimal performance.

Table 1: **Experiment for the Different Components**

| | Datasets | ETTm1 | | | | PEMS04 | | | |
| Component | In/Out | 96/96 | | 96/360 | | 96/96 | | 96/360 | |
| | Metrics | MSE | MAE | MSE | MAE | MSE | MAE | MSE | MAE |
|---|---|---|---|---|---|---|---|---|---|
| | **Adaptive** | **0.3010** | **0.3411** | **0.4240** | **0.4229** | **0.2574** | **0.3329** | **0.3265** | **0.3376** |
| **Down sampling method** | Avg | 0.3268 | 0.3571 | 0.5088 | 0.5186 | 0.3179 | 0.3807 | 0.3928 | 0.4578 |
| | Max | 0.3272 | 0.3707 | 0.5289 | 0.5158 | 0.3191 | 0.3785 | 0.3751 | 0.4322 |
| | **3** | **0.3010** | **0.3411** | **0.4240** | **0.4229** | **0.2574** | **0.3329** | **0.3265** | **0.3376** |
| **Extraction Layers** | 2 | 0.3257 | 0.3626 | 0.4506 | 0.4261 | 0.3151 | 0.3880 | 0.3776 | 0.4309 |
| | 1 | 0.3209 | 0.3597 | 0.4427 | 0.4295 | 0.3302 | 0.3987 | 0.3692 | 0.4108 |
| | **DFT decomp** | **0.3010** | **0.3411** | **0.4240** | **0.4229** | **0.2574** | **0.3329** | **0.3265** | **0.3376** |
| **Decomp Method** | Moving Avg | 0.3262 | 0.3609 | 1.1975 | 0.7098 | 0.3189 | 0.3836 | 0.3800 | 0.4366 |
| | Global Avg | 0.3305 | 0.3703 | 0.4301 | 0.4433 | 0.3033 | 0.3878 | 0.3919 | 0.4103 |
| | **Period-Trend** | **0.3010** | **0.3411** | **0.4240** | **0.4229** | **0.2574** | **0.3329** | **0.3265** | **0.3376** |
| **Attention Mode** | Trend-Period | 0.3286 | 0.3676 | 0.4477 | 0.4360 | 0.3346 | 0.3880 | 0.3699 | 0.4643 |
| | Period | 0.3436 | 0.3632 | 0.4882 | 0.4432 | 0.4744 | 0.4518 | 0.3940 | 0.4263 |
| | Trend | 0.3321 | 0.3604 | 0.4997 | 0.4514 | 0.6085 | 0.5590 | 0.3730 | 0.4185 |
| | **All Adapt** | **0.3010** | **0.3411** | **0.4240** | **0.4229** | **0.2574** | **0.3329** | **0.3265** | **0.3376** |
| **Fusion Approach** | Random | 0.3404 | 0.3786 | 0.4387 | 0.4309 | 0.4591 | 0.4720 | 0.5133 | 0.4705 |
| | Coarsest | 0.3246 | 0.3600 | 0.4445 | 0.4383 | 0.3322 | 0.3870 | 0.4275 | 0.4322 |
| | Finest | 0.3398 | 0.3735 | 0.4594 | 0.4384 | 0.3404 | 0.3810 | 0.4224 | 0.4174 |

### 3.1.4 PARAMETRIC ANALYSIS

We analyze three key AutoMixer parameters—feature dimension ($d_{model}$), encoder layers, and decoder layers—which shape the multi-scale cross-attention framework and impact detection accuracy. Table 2 shows that $d_{model} = 32$ yields the lowest error, balancing representational power and computational efficiency to avoid underfitting or overfitting. Optimal performance occurs with 3 encoder layers and 1 decoder layer, as deeper encoders cause gradient redundancy and additional decoder layers introduce noise. These parameters optimize AutoMixer's dual cross-attention mechanism for accurate long-term safety risk detection. Experiments on Metr-LA, PEMS-BAY, PEMS03/08, and

Traffic datasets confirm AutoMixer's superior MSE and MAE performance over baselines, demonstrating robustness, strong prediction and abnormal risk detection capabilities

Table 2: **Parameter Sensitivity Analysis**

| Parameter | Datasets | Traffic | | | | Metr-la | | | |
|---|---|---|---|---|---|---|---|---|---|
| | In/Out | 96/96 | | 96/720 | | 96/96 | | 96/720 | |
| | Metrics | MSE | MAE | MSE | MAE | MSE | MAE | MSE | MAE |
| D Model | 16 | 0.4887 | 0.2802 | 0.5647 | 0.3140 | 1.2010 | 0.6773 | 1.6112 | 0.8163 |
| | 24 | 0.4827 | 0.2853 | 0.5893 | 0.3169 | 1.2300 | 0.6802 | 1.6050 | 0.8103 |
| | 32 | **0.4003** | **0.2707** | **0.4912** | **0.3079** | **1.0687** | **0.6319** | **1.5492** | **0.7593** |
| | 40 | 0.4819 | 0.2812 | 0.6181 | 0.3186 | 1.2574 | 0.6844 | 1.6462 | 0.8066 |
| | 48 | 0.4827 | 0.2746 | 0.5809 | 0.3191 | 1.2315 | 0.7062 | 1.6003 | 0.8231 |
| E Layers | 1 | 0.4830 | 0.3097 | 0.5525 | 0.3408 | 1.2577 | 0.6875 | 1.6289 | 0.8115 |
| | 2 | 0.4881 | 0.2877 | 0.5785 | 0.3376 | 1.2560 | 0.6882 | 1.5965 | 0.8319 |
| | 3 | **0.4003** | **0.2707** | **0.4912** | **0.3079** | **1.0687** | **0.6319** | **1.5492** | **0.7593** |
| | 4 | 0.4734 | 0.2850 | 0.6081 | 0.3139 | 1.2832 | 0.6825 | 1.6241 | 0.7923 |
| | 5 | 0.4840 | 0.2861 | 0.6058 | 0.3124 | 1.2138 | 0.6864 | 1.5665 | 0.8436 |
| D Layers | 1 | **0.4003** | **0.2707** | **0.4912** | **0.3079** | **1.0687** | **0.6319** | **1.5492** | **0.7593** |
| | 2 | 0.4997 | 0.2802 | 0.5588 | 0.3442 | 1.2225 | 0.7028 | 1.6801 | 0.8224 |
| | 3 | 0.4706 | 0.2775 | 0.5486 | 0.3395 | 1.2241 | 0.7030 | 1.6614 | 0.8335 |
| | 4 | 0.4887 | 0.2780 | 0.5775 | 0.3083 | 1.2243 | 0.7026 | 1.6726 | 0.8354 |
| | 5 | 0.4796 | 0.2782 | 0.5528 | 0.3370 | 1.2227 | 0.7028 | 1.6797 | 0.8352 |

### 3.1.5 IMPACT OF THE SPATIAL NODES SCALE

We evaluate AutoMixer's detection accuracy and efficiency on large-scale traffic datasets from Los Angeles (1,500 nodes) and London (3,000 nodes). As shown in Table 3, AutoMixer consistently outperforms baseline models, demonstrating strong scalability and prediction capability in high-dimensional safety scenarios. These results validate the effectiveness of the dual cross-attention framework in handling complex, large-scale safety data, and highlight its potential for real-world deployment in expansive sensor networks.

Table 3: **Impact of the Spatial Nodes Scale**

| Dataset | Model | MSE | | | | MAE | | | |
|---|---|---|---|---|---|---|---|---|---|
| | In/Out | 96/96 | 96/192 | 96/360 | 96/720 | 96/96 | 96/192 | 96/360 | 96/720 |
| Los Angeles | AutoMixer | **0.4502** | **0.4974** | **0.5179** | 0.5472 | **0.4717** | **0.4858** | **0.4923** | 0.5370 |
| | TimeMixer | 0.5332 | 0.6429 | 0.6851 | 0.7209 | 0.5156 | 0.5802 | 0.6108 | 0.6322 |
| 1500 | PatchTST | 0.4960 | 0.5249 | 0.5327 | **0.5427** | 0.5325 | 0.5297 | 0.5529 | **0.5283** |
| | Informer | 1.2412 | 1.2049 | 1.2018 | NA | 0.8915 | 0.8790 | 0.8779 | NA |
| | AutoFormer | 1.6249 | 1.9430 | 1.9625 | NA | 1.0040 | 1.1211 | 1.1356 | NA |
| | Transformer | 1.2143 | 1.1892 | 1.1987 | NA | 0.8807 | 0.8730 | 0.8772 | NA |
| | DLinear | 0.7949 | 0.7315 | 0.6793 | NA | 0.7013 | 0.6668 | 0.6412 | NA |
| London | AutoMixer | **40.1655** | **50.8891** | 62.4224 | 67.9954 | **1.0557** | **1.1121** | **1.1632** | **1.1838** |
| | TimeMixer | 41.1397 | 52.6079 | **61.0205** | 68.5747 | 1.0943 | 1.1153 | 1.1706 | 1.2265 |
| 3000 | PatchTST | 44.5709 | 58.8722 | 64.8451 | **67.8126** | 1.1165 | 1.1483 | 1.1700 | 1.1935 |
| | Informer | 64608.0230 | 64975.4425 | 65342.8438 | NA | 169.2070 | 169.4406 | 169.6610 | NA |
| | AutoFormer | 42715.5390 | 47358.3040 | 52001.0660 | NA | 133.6510 | 141.7786 | 149.8980 | NA |
| | Transformer | 63942.9530 | 64233.2847 | 64523.6050 | NA | 168.7230 | 168.9973 | 169.2630 | NA |
| | DLinear | 68869.4600 | 70105.8634 | 71342.2570 | NA | 167.3190 | 168.1663 | 169.0000 | NA |

### 3.1.6 EFFICIENCY EXPERIMENT

Table 4: **Efficiency Experiment**

| Methods | Model size | Training time | Inference Time | MSE | MAE |
|---|---|---|---|---|---|
| AutoMixer | 698865 | 80.2078 | **0.0168** | **1.5492** | **0.7593** |
| TimeMixer | 230329 | 96.2897 | 0.0169 | 1.7518 | 0.8267 |
| PatchTST | 1106144 | 68.3569 | 0.1747 | 1.5552 | 0.7952 |
| Informer | 12045007 | 67.8122 | 0.3347 | 1.5734 | 0.8387 |
| AutoFormer | 11560143 | 77.0666 | 0.7660 | 2.0529 | 0.9651 |
| DLinear | **139680** | **29.5118** | 0.2421 | 1.6152 | 0.8147 |
| Transformer | 11257039 | 88.0850 | 0.4212 | 1.8633 | 0.8922 |

This experiment evaluates the detection performance and computational efficiency of the model. As shown in Table 4, although AutoMixer does not have the lowest parameter count or the short-

est training time, its design avoids spatial modules and complex convolutions by utilizing cross-attention post-spatiotemporal decomposition. This results in faster inference and higher detection accuracy. This balance of efficiency and effectiveness positions AutoMixer as one of the fastest and most capable models for anomaly and risk detection available today. Additionally, we compared AutoMixer's GPU memory usage and training speed with other models, as shown in Fig. 3. AutoMixer demonstrates exceptional efficiency, effectively utilizing GPU resources and maintaining fast runtime across sequence lengths from 96 to 720. Its lightweight design, which excludes spatial modules, significantly increases training speed while maintaining excellent performance in security risk detection.

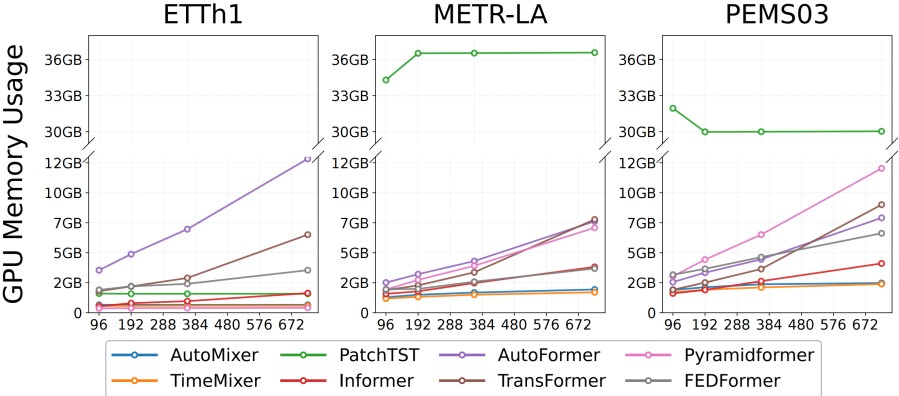

Figure 3: Memory Efficiency Analysis Across Different Time Series Lengths

### 3.1.7 DATA NOISE ANALYSIS

This experiment evaluated AutoMixer's robustness to noisy sensor data, simulating real-world conditions with data loss or errors. Noise levels of 5%, 10%, 20%, and 40% were added to the training set, while testing was conducted on the original clean data. As shown in Appendix A.2.6 Table 7, prediction metrics remained largely stable despite noise introduction. Notably, on the Met-la dataset, the model trained with noise outperformed the noise-free baseline across all forecast horizons. These results demonstrate the strong resilience of the dual cross-attention framework in handling noisy spatiotemporal sequences, underscoring its practical value for real safety applications.

### 3.1.8 ZERO-SHOT STUDY

We evaluated AutoMixer's zero-shot generalization by applying models trained on Electricity and Metr-la datasets to the ETT series. As shown in Appendix A.2.7 Table 8, AutoMixer outperformed baseline models in accuracy for certain forecast horizons. Its frequency-domain adaptive decomposition and dual cross-attention framework effectively capture multi-scale feature patterns, enabling strong cross-dataset transfer and zero-shot prediction. Compared to TimeMixer's decomposable mixing, AutoMixer's dynamic coupling dual-attention architecture improves adaptability, reducing errors by 8% in zero-shot scenarios. These results highlight the practical value of AutoMixer for real-world safety detection, particularly in data-scarce or missing-data conditions.

## 4 CONCLUSION

The proposed AutoMixer model leverages frequency-domain adaptive decomposition of spatio-temporal sequences, dynamic coupling weights, a dual-attention analysis framework, and multi-resolution dynamic coupling modules to achieve efficient anomaly and risk detection. Evaluated on both traffic safety and industrial safety datasets, AutoMixer consistently outperforms baselines, reducing prediction errors by over 4.2%. It demonstrates exceptional scalability on large-scale data with more than 3000 spatial nodes. The dual cross-attention mechanism effectively balances accuracy and training efficiency, maintaining robust performance even with noisy inputs. These strengths position AutoMixer as a powerful tool for real-time detection applications in domains such as accident prediction and equipment monitoring.

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

## LLMs Usage Disclosure

The authors disclose that Large Language Models (LLMs) were used only to aid or polish writing.

## A Appendix

### A.1 Related Work

#### A.1.1 Traditional Time Series Forecasting Methods

Traditional time series forecasting relies on classical statistical models and machine learning algorithms with simple structures, limited parameters, and strong stability and interpretability. These approaches face challenges in handling complex nonlinear or high-dimensional spatiotemporal data relevant to traffic safety and Industry 5.0 system reliability. Feature-based methods manually extract relevant features and apply regression models for prediction. For instance, STI-WSFM Ohashi & Torgo (2012) uses spatially weighted historical averages and temporal window statistics combined with regression trees or SVMs to analyze accident patterns. FGAQFM Zheng et al. (2015) integrates local global and abrupt spatiotemporal features through linear regression and neural networks, but depends heavily on domain expertise, exhibits poor generalization, and is sensitive to noise in safety-critical datasets. More recent spatiotemporal selective state-space models (ST-SSMs) Shao et al. (2024) improve accuracy by adaptively focusing on important spatiotemporal patterns in traffic and industrial safety data.

Classical ARIMA models Kontopoulou et al. (2023) perform well on stationary series via autoregression, differencing, and moving average components for accident and anomaly prediction. Gaussian process methods, as nonparametric Bayesian models, capture correlations through kernel functions and provide uncertainty quantification for equipment health monitoring. For example, Senanayake et al. Senanayake et al. (2016) combine kernels with variational inference to enhance scalability in industrial safety applications. These methods excel in interpretability but suffer from high computational complexity and kernel sensitivity, limiting their applicability to large-scale traffic safety and Industry 5.0 datasets.

#### A.1.2 Spatio-temporal Deep Learning Models

With advances in deep learning and computational power, spatiotemporal models increasingly incorporate graph convolutional networks (GCNs) to capture both spatial and temporal dependencies, making them suitable for complex prediction tasks in traffic safety and Industry 5.0 system reliability. For example, Hussain et al. Hussain et al. (2021) employ GRUs to model temporal dynamics in accident detection and optimize hyperparameters, STGCN Yu et al. (2017) combines GCNs and RNNs to address graph-structured spatiotemporal dependencies in road networks, and Graph-WaveNet Wu et al. (2019) integrates temporal convolutions with adaptive GCNs, enhancing interpretability and achieving strong performance in anomaly detection.

However, these models struggle to scale efficiently to large datasets, as their computational complexity increases sharply with network size, resulting in decreased efficiency in large-scale traffic or industrial sensor networks. For instance, DCRNN Zhou & Yu (2023) models spatial dependencies via diffusion processes on graphs coupled with RNNs for temporal modeling of hazardous road conditions, but it becomes computationally expensive in large-node networks like urban traffic systems. Similarly, TCGCN Wang et al. (2024a) enhances accuracy by incorporating multidimensional cross-attention and spatiotemporal graph convolutions for equipment fault detection, but it demands substantial computational resources. Systematic reviews indicate that although these approaches excel in handling non-Euclidean spatial structures in safety applications, their scalability limitations pose significant challenges for real-time Industry 5.0 and traffic safety tasks.

#### A.1.3 Large Language Models of Spatio-temporal Sequence

Recent large-scale models feature massive parameter counts and complex architectures. They undergo self-supervised pre-training on vast amounts of unlabeled data, followed by fine-tuning with minimal labeled samples for traffic safety and Industry 5.0 applications. Adhering to the "scaling

law," these models leverage increasingly deep Transformer structures Brown et al. (2020); Touvron et al. (2023) and advanced fine-tuning techniques such as chain-of-thought Wei et al. (2022) and tree-of-thought Yao et al. (2024). This paradigm enables the extraction of precise inductive biases from large datasets, unveiling strong reasoning abilities and emergent phenomena in accident and anomaly prediction. For instance, Time-LLM Jin et al. (2024) utilizes a reprogramming module to align time series with text prototypes, enabling effective predictions without modifying the underlying language model backbone for traffic safety tasks. Timer Liu et al. (2024b) adopts a GPT-like architecture Brown et al. (2020), pre-trained on a dataset containing 10 billion time points, and unifies heterogeneous sequences into generative tasks covering forecasting, imputation, and anomaly detection in industrial systems. ForecastPFN Dooley et al. (2024) demonstrates strong capabilities in cross-domain transfer, few-shot, and zero-shot forecasting for safety-critical scenarios.

Large models also show promising potential for spatiotemporal sequence prediction in Industry 5.0 and traffic safety. ST-LLM Liu et al. (2024a) treats spatiotemporal points as tokens and incorporates convolutional layers for effective information fusion in accident detection. It employs partial freezing of attention layers to capture global dependencies, supporting few-shot and zero-shot prediction scenarios in industrial monitoring. PatchTST Nie et al. (2022) achieves efficient long-term forecasting by employing patching strategies that reduce computational cost while maintaining accuracy in safety applications. UrbanGPT Li et al. (2024) combines spatiotemporal dependency encoders with instruction fine-tuning to model traffic accident dynamics in data-scarce environments, addressing applications such as crash prediction and equipment failure detection.

Despite these advances, large models face challenges, including the need for extensive high-quality data, substantial computational demands, limited interpretability, and instability when confronted with varying spatiotemporal scales or abrupt data shifts in Industry 5.0 and traffic safety tasks, constraining their widespread adoption in practical prediction scenarios.

### A.1.4 SPATIO-TEMPORAL SEQUENCE ANALYSIS METHODS WITHOUT SPATIAL MODULES

Although current spatiotemporal big data models achieve promising results, many focus exclusively on temporal analysis to improve efficiency in traffic safety and Industry 5.0 applications. By omitting graph-based spatial operations, these models reduce computational overhead, making them suitable for high-frequency, large-node datasets like accident records or industrial sensor logs. For example, Crossformer Zhang & Yan (2023) leverages dimension-segmented embeddings and dual-stage attention to capture cross-temporal variable dependencies in traffic accident data. TimeMixer Wang et al. (2024b) is a decomposable multiscale mixing model that excels at time series forecasting by decomposing sequences into trend and seasonal components for equipment anomaly detection. Informer Zhou et al. (2021) introduces ProbSparse self-attention to optimize long-sequence forecasting, effectively addressing memory and computation constraints in safety-critical tasks. AutoFormer Wu et al. (2021) employs a decomposition architecture with Auto-Correlation mechanisms to discover dependencies and merge sub-series in traffic safety datasets. FedFormer Zhou et al. (2022) integrates frequency-domain representations with attention to enhance pattern extraction in industrial sensor logs. DLinear Zeng et al. (2023) proposes a simple decomposition-linear model that isolates trend and seasonal components, achieving competitive performance with minimal parameters in Industry 5.0 applications.

However, these approaches risk losing critical information by ignoring spatial dependencies, limiting their ability to capture complex global dynamics and weakening robustness in scenarios where spatiotemporal interactions are tightly coupled, such as intersection accidents or multi-sensor anomalies in industrial systems, potentially reducing forecasting accuracy under challenging real-world safety conditions.

### A.2 ADDITIONAL EXPERIMENTAL SETTINGS

We provide a detailed description of the baseline models employed for comparative evaluation, and outline the primary metrics used to assess the performance and effectiveness of the proposed method.

### A.2.1 ADDITIONAL EXPERIMENTAL BASELINES

We compare a diverse set of baseline methods for long-term spatio-temporal sequence prediction. These baselines include TimeMixer, Informer, AutoFormer, FedFormer, PyraFormer, DLinear, and the large-scale model slicing approach PatchTST. Together, these models represent a broad spectrum of forecasting paradigms, encompassing spatio-temporal sequence modeling, large-scale attention-based architectures, and efficient linear decomposition techniques. This comprehensive comparison enables a robust evaluation of the proposed model's performance across different forecasting strategies.

### A.2.2 ADDITIONAL EXPERIMENTAL EVALUATION METRICS

We adopt Mean Squared Error (MSE) and Mean Absolute Error (MAE) as the primary evaluation metrics. MSE penalizes larger errors by squaring the difference between predictions and ground truth, thereby emphasizing outliers. In contrast, MAE measures the average magnitude of errors in a more robust and interpretable manner by taking absolute differences.

$$\text{MSE} = \frac{1}{QN} \sum_{t=1}^{Q} \left\| Y_t - \hat{Y}_t \right\|_2^2, \tag{18}$$

$$\text{MAE} = \frac{1}{QN} \sum_{t=1}^{Q} \left| Y_t - \hat{Y}_t \right|, \tag{19}$$

where $Q$ denotes the prediction horizon, $N$ is the number of predicted variables, $Y_t$ is the ground truth at time step $t$, and $\hat{Y}_t$ is the corresponding prediction.

### A.2.3 ADDITIONAL EXPERIMENTAL MODEL IMPLEMENTATION DETAILS

All experiments were conducted on an RTX 4090 GPU cluster and implemented using Python 3.8 with PyTorch 2.1.1. For a fair comparison, all baseline models adopted the Huber loss function and a consistent batch size of 64. The datasets were split into training, validation, and test sets with a ratio of 8:1:1. Each model was trained for 100 epochs, and the version yielding the best performance on the validation set was selected for final evaluation on the test set.

## A.2.4 INDUSTRIAL PRODUCTION SAFETY DETECTION DATAS EXPERIMENTS

Table 5: **Performance for Industrial Production Safety Detection Datas**

| Dataset | Metrics
Input
Output | MSE
336
96 | 336
192 | 336
336 | 336
720 | MAE
336
96 | 336
192 | 336
336 | 336
720 |
|---|---|---|---|---|---|---|---|---|---|
| ETTm1 | AutoMixer | **0.3010** | **0.3327** | **0.3896** | **0.4240** | 0.3411 | **0.3575** | **0.3742** | **0.4229** |
| | TimeMixer | 0.3367 | 0.3701 | 0.4030 | 0.4525 | 0.3740 | 0.3893 | 0.4129 | 0.4403 |
| | PatchTST | 0.3050 | 0.3408 | 0.3905 | 0.4254 | **0.3220** | 0.3604 | 0.3840 | 0.4253 |
| | Informer | 0.6909 | 0.7288 | 1.0599 | 0.9609 | 0.5991 | 0.6233 | 0.8031 | 0.7248 |
| | AutoFormer | 0.6689 | 0.6935 | 0.6536 | 0.7382 | 0.5407 | 0.5532 | 0.5447 | 0.5761 |
| | FedFormer | 0.3644 | 0.4081 | 0.4419 | 0.4923 | 0.4141 | 0.4366 | 0.4568 | 0.4813 |
| | PyraFormer | 0.5191 | 0.5736 | 0.8973 | 0.8492 | 0.4899 | 0.5410 | 0.7119 | 0.6909 |
| | DLinear | 0.3174 | 0.3407 | 0.3924 | 0.4312 | 0.3642 | 0.3744 | 0.3921 | 0.4318 |
| ETTm2 | AutoMixer | **0.1721** | 0.2371 | 0.3009 | **0.3695** | 0.2518 | 0.3154 | 0.3018 | 0.3949 |
| | TimeMixer | 0.1773 | **0.2357** | 0.3097 | 0.3988 | 0.2630 | 0.3190 | 0.3457 | 0.3997 |
| | PatchTST | 0.1739 | 0.2373 | **0.2962** | 0.3712 | 0.2546 | 0.3165 | **0.2918** | 0.3966 |
| | Informer | 0.3345 | 0.6651 | 1.4156 | 2.0935 | 0.4415 | 0.6310 | 0.9317 | 1.1553 |
| | AutoFormer | 0.2259 | 0.2868 | 0.3412 | 0.4291 | 0.3143 | 0.3502 | 0.3826 | 0.4272 |
| | FedFormer | 0.1972 | 0.2626 | 0.3265 | 0.4400 | 0.2868 | 0.3270 | 0.3651 | 0.4284 |
| | PyraFormer | 0.4779 | 0.6806 | 1.1549 | 4.4416 | 0.5370 | 0.6443 | 0.8311 | 1.6405 |
| | DLinear | 0.1755 | 0.2382 | 0.3014 | 0.4324 | 0.2767 | 0.3169 | 0.3710 | 0.4445 |
| ETTh1 | AutoMixer | **0.3862** | 0.4451 | 0.4485 | 0.4992 | **0.4005** | 0.4515 | **0.4308** | 0.4915 |
| | TimeMixer | 0.3896 | 0.4389 | 0.4895 | 0.5311 | 0.4050 | 0.4522 | 0.4384 | 0.4925 |
| | PatchTST | 0.3999 | **0.4280** | 0.4643 | 0.5077 | 0.4060 | **0.4280** | 0.4439 | **0.4810** |
| | Informer | 0.8457 | 0.9903 | 1.0832 | 1.1254 | 0.6862 | 0.7438 | 0.8028 | 0.8281 |
| | AutoFormer | 0.6406 | 0.6166 | 0.5774 | 0.6632 | 0.5558 | 0.5498 | 0.5347 | 0.5773 |
| | FedFormer | 0.3872 | 0.4477 | 0.4582 | **0.4865** | 0.4262 | 0.4519 | 0.4655 | 0.4934 |
| | PyraFormer | 0.6770 | 0.8104 | 0.8832 | 0.9985 | 0.6185 | 0.6954 | 0.7310 | 0.8034 |
| | DLinear | 0.3871 | 0.4543 | **0.4477** | 0.5066 | 0.4109 | 0.4534 | 0.4546 | 0.5187 |
| ETTh2 | AutoMixer | **0.2813** | **0.3705** | 0.4048 | 0.4405 | 0.3343 | **0.3831** | 0.3956 | 0.4572 |
| | TimeMixer | 0.2984 | 0.3772 | 0.4170 | 0.4546 | 0.3481 | 0.3986 | 0.4309 | 0.4578 |
| | PatchTST | 0.3046 | 0.3759 | **0.3949** | **0.4315** | 0.3477 | 0.3968 | 0.4156 | **0.4552** |
| | Informer | 2.5561 | 2.4144 | 4.7914 | 4.1613 | 1.3156 | 1.2398 | 1.8228 | 1.7879 |
| | AutoFormer | 0.4091 | 0.4688 | 0.4498 | 0.4505 | 0.4543 | 0.4796 | 0.4751 | 0.4816 |
| | FedFormer | 0.3506 | 0.4510 | 0.5034 | 0.4714 | 0.3953 | 0.4531 | 0.4912 | 0.4790 |
| | PyraFormer | 1.4491 | 4.7154 | 4.4482 | 4.2851 | 0.9361 | 1.7535 | 1.7457 | 1.7980 |
| | DLinear | 0.2938 | 0.3982 | 0.4635 | 0.7387 | **0.2972** | 0.3990 | 0.4688 | 0.7419 |
| Electricity | AutoMixer | 0.1525 | 0.1620 | **0.1735** | **0.2147** | **0.2442** | **0.2509** | 0.2734 | **0.3027** |
| | TimeMixer | 0.1529 | **0.1605** | 0.1815 | 0.2316 | 0.2521 | 0.2830 | **0.2591** | 0.3164 |
| | PatchTST | 0.1567 | 0.1750 | 0.1899 | 0.2319 | 0.2486 | 0.2714 | 0.2826 | 0.3181 |
| | Informer | **0.1515** | 0.1663 | 0.1906 | 0.2346 | 0.2464 | 0.2582 | 0.2847 | 0.3285 |
| | AutoFormer | 0.3488 | 0.3621 | 0.3675 | 0.4014 | 0.4246 | 0.4427 | 0.4435 | 0.4611 |
| | FedFormer | 0.7024 | 0.6137 | 0.6816 | 0.6686 | 0.6481 | 0.6082 | 0.6428 | 0.6351 |
| | PyraFormer | 0.2059 | 0.2180 | 0.2310 | 0.2772 | 0.3199 | 0.3315 | 0.3424 | 0.3764 |
| | DLinear | 0.3817 | 0.3769 | 0.3721 | 0.3739 | 0.4446 | 0.4421 | 0.4404 | 0.4461 |
| Weather | AutoMixer | **0.1590** | **0.2031** | **0.2678** | **0.3349** | 0.2019 | 0.2940 | **0.2441** | **0.3373** |
| | TimeMixer | 0.1682 | 0.2123 | 0.2745 | 0.3450 | 0.2127 | 0.2556 | 0.2968 | 0.3459 |
| | PatchTST | 0.1929 | 0.2227 | 0.2790 | 0.3385 | **0.1771** | **0.2199** | 0.2692 | 0.3422 |
| | Informer | 0.5717 | 0.3810 | 0.4171 | 0.4516 | 0.5394 | 0.4383 | 0.4588 | 0.4840 |
| | AutoFormer | 0.2534 | 0.2974 | 0.3736 | 0.3970 | 0.3197 | 0.3518 | 0.4047 | 0.4022 |
| | FedFormer | 0.2299 | 0.4494 | 0.3588 | 0.4108 | 0.3070 | 0.4649 | 0.3989 | 0.4253 |
| | PyraFormer | 0.6371 | 0.4875 | 0.9924 | 1.4466 | 0.5680 | 0.5080 | 0.7428 | 0.9492 |
| | DLinear | 0.1856 | 0.2245 | 0.2706 | 0.3376 | 0.2394 | 0.2805 | 0.3243 | 0.3751 |

### A.2.5 ADDITIONAL TRAFFIC OPERATION SAFETY DETECTION DATAS EXPERIMENTS

Table 6: **Performance Metrics for Traffic Operation Safety Detection Datas**

| Dataset | Metrics Input Output | MSE 336 96 | 336 192 | 336 336 | 336 720 | MAE 336 96 | 336 192 | 336 336 | 336 720 |
|---|---|---|---|---|---|---|---|---|---|
| Metr-la | AutoMixer | **1.0687** | **1.2114** | **1.3250** | 1.5492 | **0.6319** | **0.7101** | **0.7047** | **0.7593** |
| | TimeMixer | 1.2330 | 1.4060 | 1.4426 | 1.7313 | 0.6759 | 0.7625 | 0.7849 | 0.8497 |
| | PatchTST | 1.0731 | 1.2165 | 1.3273 | 1.5552 | 0.6628 | 0.7150 | 0.7499 | 0.7952 |
| | Informer | 1.4043 | 1.4592 | 1.4899 | 1.5734 | 0.8127 | 0.8230 | 0.8284 | 0.8387 |
| | AutoFormer | 1.5828 | 1.6943 | 1.7067 | 2.0529 | 0.8690 | 0.8873 | 0.8689 | 0.9651 |
| | FedFormer | 1.3174 | 1.4380 | 1.9610 | 1.7121 | 0.7314 | 0.7869 | 0.9057 | 0.8541 |
| | PyraFormer | 1.2159 | 1.3383 | 1.4584 | 1.6016 | 0.6612 | 0.7198 | 0.7230 | 0.8071 |
| | DLinear | 1.0933 | 1.2179 | 1.3291 | **1.4650** | 0.7181 | 0.7580 | 0.7834 | 0.8252 |
| Pems-bay | AutoMixer | 0.5856 | 0.6060 | **0.6330** | 0.9879 | 0.3729 | 0.4018 | 0.4213 | 0.4439 |
| | TimeMixer | 0.8586 | 0.9505 | 0.8789 | 1.0009 | 0.5076 | 0.5295 | 0.5042 | 0.5609 |
| | PatchTST | 0.6218 | 0.6513 | 0.7068 | 0.9989 | 0.3863 | 0.4037 | **0.4202** | **0.4392** |
| | Informer | 0.9026 | 0.9472 | 0.9694 | 0.9891 | 0.5490 | 0.5639 | 0.5716 | 0.5966 |
| | AutoFormer | 1.5647 | 1.6569 | 1.4609 | **0.9871** | 0.8000 | 0.8409 | 0.7599 | 0.8942 |
| | FedFormer | 0.6424 | 0.6217 | 0.6330 | 0.9884 | 0.4411 | 0.4329 | 0.4387 | 0.4451 |
| | PyraFormer | **0.5425** | **0.5845** | 0.6402 | 1.0508 | **0.3679** | **0.3902** | 0.4381 | 0.5245 |
| | DLinear | 0.6458 | 0.6814 | 0.7298 | 0.9993 | 0.4218 | 0.4391 | 0.4542 | 0.4709 |
| Pems03 | AutoMixer | 0.1555 | 0.1704 | 0.1811 | **0.2117** | **0.2330** | 0.2713 | **0.2734** | **0.2920** |
| | TimeMixer | 0.2774 | 0.3246 | 0.3129 | 0.3568 | 0.3464 | 0.3726 | 0.3653 | 0.3943 |
| | PatchTST | **0.1443** | **0.1702** | **0.1888** | 0.2412 | 0.2341 | **0.2676** | 0.2774 | 0.3121 |
| | Informer | 0.5702 | 0.5858 | 0.6047 | 0.7138 | 0.6012 | 0.6090 | 0.6206 | 0.6816 |
| | AutoFormer | 1.9999 | 1.9561 | 1.5149 | 1.6332 | 1.2185 | 1.1937 | 0.9964 | 1.0437 |
| | FedFormer | 0.1890 | 0.1915 | 0.1959 | 0.2386 | 0.2881 | 0.2972 | 0.2951 | 0.3324 |
| | PyraFormer | 0.1686 | 0.1723 | 0.1810 | 0.2157 | 0.2666 | 0.2721 | 0.2741 | 0.2996 |
| | DLinear | 0.2425 | 0.2488 | 0.2698 | 0.3201 | 0.3598 | 0.3631 | 0.3671 | 0.4216 |
| Pems08 | AutoMixer | **0.4554** | **0.4705** | 0.4983 | 0.5371 | 0.3522 | **0.3562** | **0.3543** | **0.3793** |
| | TimeMixer | 0.4844 | 0.3381 | 0.5319 | 0.6530 | 0.4004 | 0.4254 | 0.4788 | 0.5379 |
| | PatchTST | 0.4673 | 0.5812 | 0.6327 | 0.7190 | 0.3622 | 0.4058 | 0.4424 | 0.4892 |
| | Informer | 1.0769 | 1.1042 | 1.1070 | 1.1987 | 0.7303 | 0.7430 | 0.7439 | 0.7860 |
| | AutoFormer | 2.1981 | 2.1711 | 1.8127 | 1.9483 | 1.1551 | 1.1363 | 0.9873 | 1.0312 |
| | FedFormer | 0.5795 | 0.5973 | 0.6052 | 0.6140 | 0.4174 | 0.4096 | 0.4081 | 0.4216 |
| | PyraFormer | 0.4645 | 0.4791 | **0.4880** | **0.5070** | **0.3474** | 0.3581 | 0.3618 | 0.3692 |
| | DLinear | 0.6422 | 0.6763 | 0.7009 | 0.7954 | 0.4791 | 0.4898 | 0.5117 | 0.5575 |
| Traffic | AutoMixer | 0.4003 | 0.4256 | 0.4240 | 0.4912 | 0.2707 | 0.2743 | 0.2967 | **0.3079** |
| | TimeMixer | 0.4787 | 0.4773 | 0.4775 | 0.5481 | 0.2828 | 0.2824 | 0.2992 | 0.3120 |
| | PatchTST | **0.3742** | **0.3946** | **0.4149** | 0.4928 | **0.2658** | **0.2657** | **0.2785** | 0.3260 |
| | Informer | 1.1676 | 1.3539 | 1.4088 | 1.4720 | 0.6369 | 0.7322 | 0.7700 | 0.8028 |
| | AutoFormer | 0.7078 | 0.7203 | 0.7313 | 1.5136 | 0.4493 | 0.4535 | 0.4631 | 0.8312 |
| | FedFormer | 0.5749 | 0.6139 | 0.6213 | 0.6303 | 0.3576 | 0.3811 | 0.3800 | 0.3829 |
| | PyraFormer | 0.8704 | 0.8704 | 0.8763 | 0.8998 | 0.4692 | 0.4672 | 0.4672 | 0.4738 |
| | DLinear | 0.4149 | 0.4339 | 0.4432 | 0.4763 | 0.2971 | 0.2988 | 0.3105 | 0.3236 |

### A.2.6 DATA NOISE ANALYSIS EXPERIMENTS

Table 7: **Performance Of The Spatial Noise**

| Dataset | Metrics | MSE | | | | MAE | | | |
|---|---|---|---|---|---|---|---|---|---|
| | In/Out | 96/96 | 96/192 | 96/360 | 96/720 | 96/96 | 96/192 | 96/360 | 96/720 |
| Etth1 | Original data | 0.4351 | **0.4432** | 0.6674 | 0.4351 | **0.4336** | **0.4417** | 0.5502 | **0.4336** |
| | 5% Noise | 0.4359 | **0.4449** | 0.6679 | 0.4352 | 0.4347 | 0.4435 | 0.5510 | 0.4349 |
| | 10% Noise | 0.4360 | 0.4452 | 0.6682 | 0.4368 | 0.4357 | 0.4440 | 0.5518 | 0.4367 |
| | 20% Noise | **0.4218** | 0.4871 | **0.4771** | **0.4218** | 0.4547 | 0.4953 | **0.4858** | 0.4547 |
| | 40% Noise | 0.4498 | 0.4770 | 0.4904 | 0.4498 | 0.4802 | 0.5030 | 0.5100 | 0.4802 |
| Metr-la | Original data | 1.0687 | 1.2114 | 1.3250 | 1.5492 | 0.6319 | 0.7101 | 0.7047 | 0.7593 |
| | 5% Noise | 1.0742 | 1.2285 | 1.2831 | 1.4113 | 0.6427 | 0.7093 | 0.7363 | 0.7659 |
| | 10% Noise | 1.0347 | 1.1640 | 1.2234 | 1.3254 | 0.6399 | 0.6877 | 0.7156 | 0.7384 |
| | 20% Noise | 0.9559 | 0.8585 | 1.1020 | 1.1766 | 0.6165 | 0.6492 | 0.6699 | 0.6793 |
| | 40% Noise | **0.7703** | **0.8313** | **0.8621** | **0.9069** | **0.5334** | **0.5407** | **0.5378** | **0.5386** |
| Pems03 | Original data | **0.1555** | **0.1704** | **0.1811** | **0.2117** | **0.2330** | **0.2713** | **0.2734** | **0.2920** |
| | 5% Noise | 0.2534 | 0.2930 | 0.2860 | 0.3155 | 0.3486 | 0.3802 | 0.3643 | 0.3941 |
| | 10% Noise | 0.2712 | 0.2919 | 0.2925 | 0.3199 | 0.3668 | 0.3841 | 0.3794 | 0.4056 |
| | 20% Noise | 0.3095 | 0.3284 | 0.3272 | 0.3470 | 0.4085 | 0.4238 | 0.4167 | 0.4354 |
| | 40% Noise | 0.3660 | 0.3705 | 0.3698 | 0.3744 | 0.4591 | 0.4637 | 0.4619 | 0.4664 |

### A.2.7 ZERO-SHOT STUDY EXPERIMENTS

Table 8: **Zero-Shot Performance Comparison**

| Dataset | Metrics | MSE | | | | MAE | | | |
|---|---|---|---|---|---|---|---|---|---|
| | In/Out | 96/96 | 96/192 | 96/360 | 96/720 | 96/96 | 96/192 | 96/360 | 96/720 |
| Ettm2 | AutoMixer | 0.1721 | 0.2371 | 0.3009 | 0.3695 | 0.2518 | 0.3154 | 0.3018 | 0.3949 |
| | TimeMixer | 0.1773 | 0.2357 | 0.3097 | 0.3988 | 0.2630 | 0.3190 | 0.3457 | 0.3997 |
| | PatchTST | 0.1739 | 0.2373 | 0.2962 | 0.3712 | 0.2546 | 0.3165 | 0.2918 | 0.3966 |
| | Informer | 0.3345 | 0.6651 | 1.4156 | 2.0935 | 0.4415 | 0.6310 | 0.9317 | 1.1553 |
| | AutoFormer | 0.2259 | 0.2868 | 0.3412 | 0.4291 | 0.3143 | 0.3502 | 0.3826 | 0.4272 |
| | (0-Shot of Electricity) | 0.2263 | 0.2825 | 0.3933 | 0.4707 | 0.3120 | 0.3430 | 0.4020 | 0.4481 |
| | (0-Shot of Metr-la) | 0.2368 | 0.2873 | 0.3590 | 0.4398 | 0.3194 | 0.3440 | 0.3871 | 0.4270 |
| Etth2 | AutoMixer | 0.2813 | 0.3705 | 0.4048 | 0.4405 | 0.3343 | 0.3831 | 0.3956 | 0.4572 |
| | TimeMixer | 0.2984 | 0.3772 | 0.4170 | 0.4546 | 0.3481 | 0.3986 | 0.4309 | 0.4578 |
| | PatchTST | 0.3046 | 0.3759 | 0.3949 | 0.4315 | 0.3477 | 0.3968 | 0.4156 | 0.4552 |
| | Informer | 2.5561 | 2.4144 | 4.7914 | 4.1613 | 1.3156 | 1.2398 | 1.8228 | 1.7879 |
| | AutoFormer | 0.4091 | 0.4688 | 0.4498 | 0.4505 | 0.4543 | 0.4796 | 0.4751 | 0.4816 |
| | (0-Shot of Electricity) | 0.3480 | 0.4399 | 0.5418 | 0.5006 | 0.3888 | 0.4439 | 0.4983 | 0.4916 |
| | (0-Shot of Metr-la) | 0.3023 | 0.3880 | 0.4188 | 0.4233 | 0.3469 | 0.3972 | 0.4293 | 0.4420 |

