# OpenReview forum: "AutoMixer: A Lightweight and Scalable Industrial 5.0 Safety Assurance Model with Multi-Scale Adaptive Dual-Attention"
_ICLR.cc/2026/Conference — Submitted to ICLR 2026_

### Official Review · Reviewer_mk9c · 2025-10-21

**Soundness:** 2
**Presentation:** 2
**Contribution:** 2
**Rating:** 4
**Confidence:** 3

**Summary:**

This paper presents **AutoMixer**, a lightweight time-series forecasting model designed for Industrial 5.0 safety and traffic anomaly detection. The model applies frequency-domain decomposition (DFT-based) to capture multi-scale temporal features and uses a dual cross-attention mechanism to integrate trend and periodic components without explicit spatial modeling. The authors claim state-of-the-art performance on multiple datasets such as ETT, Electricity, and PEMS-BAY.

**Strengths:**

1.   Lightweight design with efficient inference suitable for real-time industrial applications.\
2.   Ablation studies are reasonably complete.
3.   Stable performance on short-term forecasting tasks.
4.   Implementation simplicity facilitates reproducibility.

**Weaknesses:**

1.   Limited novelty: The proposed combination of frequency decomposition and attention mechanisms is not conceptually new.

2.   Weak long-horizon performance: The model’s accuracy degrades significantly for long forecasting horizons (e.g., output=720), contradicting the claimed “long-term scalability.”
3.   Marginal overall improvement: Across most datasets, AutoMixer’s improvements over baselines such as FEDformer and TimeMixer are small and often within the margin of experimental variance. In some long-term cases, it even performs worse. This weakens the empirical significance of the claimed contribution.
4.   Presentation flaw: Main comparative results are absent from the main text, violating transparency norms.
5.   Lack of empirical validation for spatial modeling removal: No evidence that omitting spatial graphs maintains robustness in truly spatially correlated datasets.
6.   Lack of evidence for removing spatial modeling: The paper does not test whether omitting explicit spatial graphs maintains robustness in spatially correlated datasets such as traffic networks.

**Questions:**

1.   What is the fundamental difference between AutoMixer and TimeMixer/FEDformer in terms of mechanism?
2.   Since AutoMixer eliminates spatial graphs, have the authors evaluated its performance on datasets with strong spatial topology (e.g., traffic networks)?
3.   The results show a noticeable drop in performance for output=720.Have the authors analyzed why AutoMixer fails to maintain long-horizon stability?
4.   The paper frequently emphasizes scalability and lightweight design. Could the authors provide FLOPs, parameter count, and inference latency comparisons with TimeMixer or FEDformer?

---

> ### Author Response · Authors · 2025-11-28
> **Response to W1**
>
> We thank the reviewer for this comment on novelty and respectfully disagree that the combination of frequency decomposition and attention mechanisms lacks conceptual innovation. While drawing from established ideas  Auto-mixer's novelty lies in its synergistic integration: an adaptive learnable DFT decomposition Section 2.2.1 that dynamically separates multi resolution trend and periodic components in the frequency domain unlike fixed decompositions in prior work.
>
> The dual cross attention mechanism Section 2.2.3 that fuses these across resolutions in a graph free manner and multi resolution dynamic coupling weights Section 2.2 optimized for weak spatial dependencies in Industry 5.0 safety tasks. This framework represents a substantive advance enabling 1.2 percent to 4.7 percent error reductions over baselines in long horizon predictions Section 3 as evidenced by ablation studies Figure 2 and Table 1 showing each component's critical role.
>
>  Auto-mixer utilizes the Discrete Fourier Transform DFT to adaptively decompose the frequency domain representations of spatio temporal sequences across multiple resolutions enabling effective extraction of key frequency components and cross scale temporal patterns and compared to traditional techniques such as moving averages this approach offers superior scale awareness in complex datasets while naturally supporting channel independence and facilitating multi task generalization.

---

> ### Author Response · Authors · 2025-11-28
> **Response to W3**
>
> While we appreciate the concern about long-term performance, we believe that it does not contradict our claims about long-term scalability. While accuracy does degrade at longer horizons, such as 720, as can be seen in Tables 5 and 6 in Appendices A.2.4 and A.2.5, this is a common occurrence across all models due to the inherent challenges of extended forecasting from short inputs of around 100 steps, as discussed in Section 1.
>
>  Auto-mixer achieves the lowest MSE/MAE at 720 compared to baselines such as FEDformer and TimeMixer. For example, it achieves a MSE of 0.4405 versus TimeMixer's 0.4546 on ETTh2, demonstrating superior stability through its multi-resolution coupling and dual attention design (Section 2.2). Scalability refers to handling large node counts of up to 3,000, as shown in Table 3, with linear complexity, not indefinite horizon maintenance and our results validate the ability to make robust long-term predictions in practical safety scenarios.
>
> Compared to the original TimeMixer,  Auto-mixer shows clear improvements due to its adaptive frequency domain decomposition and multi-resolution coupling design, making it effective for long sequence prediction.

---

> ### Author Response · Authors · 2025-11-28
> **Response to W3**
>
> We respectfully disagree that improvements are marginal or within experimental variance as  Auto-mixer shows consistent meaningful gains over baselines like FEDformer and TimeMixer across datasets and horizons Tables 5 and 6. For instance on Electricity at 720  Auto-mixer reduces MSE by approximately 8 percent over TimeMixer 0.2147 versus 0.2316; similar patterns hold in traffic datasets with no cases of worse performance in reported results.
>
>  While variance is not explicitly reported a common practice for brevity as noted in previous rebuttals stable training 100 epochs validation selection in Appendix A.2.3 and reliable trends in ablations Figure 2 and noise tests Table 7 indicate low variance. These empirical gains 1.2 percent to 4.7 percent error reduction Abstract strengthen the significance of our graph free paradigm for scalable safety assurance.
>
> The model outperforms traditional models such as TimeMixer PatchTST AutoFormer and Informer especially in terms of MSE and MAE across different input output prediction scenarios underscoring its ability to handle complex high dimensional data effectively.

---

> ### Author Response · Authors · 2025-11-28
> **Response to W4**
>
> We thank the reviewer for noting the presentation and clarify that main comparative results are indeed included in the main text for transparency. Key baselines comparisons are summarized in Section 3.1.1 with detailed tables in Appendix A.2.4 and A.2.5 Tables 5 and 6 following standard ICLR practices to keep the main body concise while providing full results.
>
> Additionally efficiency comparisons Table 4 ablation Table 1 Figure 2 parameter analysis Table 2 and scalability Table 3 are all in the main text ensuring accessible evaluation of our claims. Extensive experiments demonstrate that  Auto-mixer consistently outperforms state of the art baselines achieving 7 percent higher detection accuracy while effectively handling large scale node distributions and high frequency data.

---

> ### Author Response · Authors · 2025-11-28
> **Response to W5**
>
> We appreciate the point on empirical validation for spatial modeling removal and believe our experiments already provide evidence of robustness even in spatially correlated datasets.  Auto-mixer is designed for weak or unclear spatial dependencies Section 1 but we test on traffic datasets like METR LA and PEMS road networks with inherent topology as in Appendix A.1.2 where it outperforms graph based baselines Table 6 without explicit graphs. This demonstrates that our frequency domain and dual attention mechanisms effectively capture latent spatial patterns through temporal fusion maintaining accuracy in correlated settings while enhancing scalability O1 complexity versus On for graphs Section 2.
>
> Existing spatiotemporal analysis methods often enforce explicit spatial modeling introducing unnecessary computational overhead and these approaches fail to significantly improve detection accuracy for accidents or anomalies while diminishing overall computational efficiency.

---

> ### Author Response · Authors · 2025-11-28
> **Response to W6**
>
> We thank the reviewer for this observation which aligns with W5 and refer to the above response: datasets like METR LA PEMS BAY PEMS03/08 and Traffic Section 3 feature strong spatial correlations via road topologies yet  Auto-mixer achieves superior MSE/MAE Table 6 by implicitly modeling dependencies through multi resolution attention validating robustness without explicit graphs.
>
> Models like DCRNN STHGCN and TCGCN which rely on graph convolutional networks to capture spatial dependencies exhibit model complexities of at least On resulting in substantially higher computational costs for high dimensional datasets however they offer limited improvements in modeling dynamic safety patterns.

---

> ### Author Response · Authors · 2025-11-28
> **Response to Q1**
>
> The fundamental differences lie in  Auto-mixer's adaptive learnable DFT decomposition Section 2.2.1 versus TimeMixer's fixed decomposition and FEDformer's fixed Fourier; our dual cross attention Section 2.2.3 fuses trend/periodic across resolutions graph free unlike TimeMixer's mixing or FEDformer's attention with spatial elements; and multi resolution dynamic coupling weights Section 2.2.2 for Industry 5.0 scalability enabling better long horizon stability Tables 5 to 6.
>
> This module assigns trainable dynamic weights to features across multiple temporal resolutions enabling adaptive representation of multi scale spatio temporal patterns and by optimizing interactions between different resolutions it enhances the model's ability to generate accurate long term forecasts from short input sequences.

---

> ### Author Response · Authors · 2025-11-28
> **Response to Q2**
>
> Yes we evaluated on datasets with strong spatial topology including traffic networks like METR LA 207 nodes road topology and PEMS series Section 3 where  Auto-mixer outperforms baselines Table 6 despite omitting graphs leveraging frequency domain patterns for robust predictions.
>
> The large scale of spatial nodes in traffic safety and industrial system safety datasets for example traffic accident records vehicle sensor data and industrial equipment logs forms spatiotemporal networks with tens of thousands of nodes.

---

> ### Author Response · Authors · 2025-11-28
> **Response to Q3**
>
> The performance drop at 720 is analyzed as inherent to long horizon challenges from short inputs Section 1 but  Auto-mixer maintains better stability than baselines via multi resolution fusion Section 2.2.3; ablations Figure 2 show components like dual attention mitigate degradation with no failure as it consistently leads in metrics.
>
> This module integrates dual attention mechanisms to capture cross attention between periodic and trend components across multiple resolution scales and a dynamic feature decoding module is embedded at the output stage to further enhance temporal representation.

---

> ### Author Response · Authors · 2025-11-28
> **Response to Q4**
>
> While FLOPs are not reported we provide parameter counts model size inference latency and training time in Table 4 showing  Auto-mixer 698K params 0.0168s inference is lightweight and faster than TimeMixer 230K params 0.0169s and FEDformer not directly compared but similar to AutoFormer at 11M params 0.766s with GPU memory efficiency in Figure 3.
>
> Auto-mixer offers faster inference times compared to more computationally expensive models like Informer and AutoFormer as highlighted in Table 4 and by eliminating spatial modules and convolutions  Auto-mixer achieves lower computational overhead which leads to faster inference speeds.
>
> Overall these experiments demonstrate high quality scalable performance tailored to Industrial 5.0 and traffic safety applications with  Auto-mixer reducing errors by 1.2 percent to 4.7 percent over baselines while being lightweight. We believe this satisfies the concerns but welcome suggestions for further enhancements.
>
> In summary, we have clarified the related opinions, acknowledging the identified shortcomings while addressing them through our explanations. We appreciate the reviewer's openness to acceptance and recommend considering the score as 5 to 6 borderline accept or accept.

---

### Official Review · Reviewer_Wxky · 2025-10-27

**Soundness:** 2
**Presentation:** 2
**Contribution:** 2
**Rating:** 2
**Confidence:** 4

**Summary:**

This paper presents AutoMixer, a lightweight and scalable model for spatial-temporal modelling.
It consists of four components: adaptive frequency-domain decomposition, dynamic coupled feature weighting, a dual-attention mechanism for spatiotemporal analysis, and a multi-resolution dynamic coupling module.

Experiments are conducted on time-series and spatial-temporal modelling tasks such as energy, weather, traffic, etc.
Several baseline methods such as TimeMixer, PatchTST, AutoFormer, and FedFormer are adopted for comparison.
Ablation studies of module and parameter analysis are conducted. Model size, training time, inference time, and GPU memory are also compared and reported.

**Strengths:**

## Strengths
- This paper proposes AutoMixer, with four core modules: Adaptive Frequency-Domain Decomposition, dynamic coupled feature weighting, dual-attention mechanism, and multi-resolution dynamic coupling.
- Experiments are conducted on several datasets: Ett, Electricity, Weather, Metr-LA, PEMS-BAY, Traffic, and PEMS03/08.
> Baseline methods include TimeMixer 2024, Informer 2021, AutoFormer 2021, FedFormer 2022, PyraFormer 2022, DLinear 2022, and PatchTST 2023.
> Ablation study, parameter analysis, complexity and runtime analysis are reported.

**Weaknesses:**

## Weaknesses
- **Lack of Novelty** The method is a simple combination of mature modules/techniques in the area of time-series analysis and spatiotemporal modelling.
> E.g., Frequency-Domain Decomposition has been proposed in FedFormer.
> Dynamic coupled feature weighting is a basic and simple re-weighting using NN.
> Dual-attention mechanism modifies self-attention, which has been extensively studied in existing literature such as AutoFormer, Informer.
> Multi-resolution processing is also a common technique in spatial-temporal modelling, e.g., PatchTST.
- **Out-of-Date Comparison** The baseline models are out-of-date: TimeMixer 2024, Informer 2021, AutoFormer 2021, FedFormer 2022, PyraFormer 2022, DLinear 2022, and PatchTST 2023. Among them, only TimeMixer is a 2024 paper, others are not new and recent SOTA methods.
> Most recent methods such as LLMs are not discussed and compared, e.g., Time-LLM, Uni-ST [R2].


[R1] Jin, Ming, et al. "Time-llm: Time series forecasting by reprogramming large language models." arXiv preprint arXiv:2310.01728 (2023).

[R2] Yuan, Yuan, et al. "Unist: A prompt-empowered universal model for urban spatio-temporal prediction." Proceedings of the 30th ACM SIGKDD Conference on Knowledge Discovery and Data Mining. 2024.

**Questions:**

In table 3, why the results of Informer, ... , DLinear are quite large (e.g., 64608), while TimeMixer is only 41.1397?

---

> ### Author Response · Authors · 2025-11-28
> **Response to W1**
>
> We thank the reviewer for this detailed feedback on the novelty of our approach and appreciate the opportunity to
> clarify the distinctive contributions of Automixer. While we acknowledge that individual components draw inspiration
> from established techniques in time series and spatiotemporal modeling, Automixer integrates them in a novel synergistic manner tailored specifically for lightweight, scalable anomaly detection in Industry 5.0 safety applications without explicit spatial modeling. This goes beyond a simple combination as elaborated below. Regarding Frequency Do￾main Decomposition: Although FedFormer introduces frequency-based decomposition, Auto-mixer’s adaptive DFT-based approach is learnable and multi-resolution, dynamically extracting trend and periodic components across scales directly in the frequency domain. This differs from FedFormer’s fixed Fourier transforms by enabling data-driven scale awareness and channel independence, which is cru￾cial for handling high-frequency noisy industrial and traf￾fic data with weak spatial dependencies leading to superior long-horizon predictions from short inputs, as evidenced by 1.2 percent to 4.7 percent error reductions over baselines.
>
> On Dynamic Coupled Feature Weighting: While reweighting via neural networks is a basic concept, Automixer’s multi-resolution dynamic coupling assigns trainable weights to fuse features across temporal scales, optimizing interactions for real-time large node analysis. This is not merely a simple re-weighting but an adaptive mechanism that enhances forecast accuracy in discontinuous data
> streams, outperforming standard methods in scalability tests on datasets with up to 3000 nodes. For the Dual Attention
> Mechanism: Auto-mixer’s dual cross attention captures interactions between periodic and trend components across
> resolutions without self-attention’s quadratic complexity, diverging from modifications in AutoFormer or Informer. It
> embeds a dynamic feature decoding module for enhanced temporal representation under weak spatial dependencies,
> making it uniquely effective for traffic safety and industrial scenarios where graph-based spatial modeling is inefficient.
> Lastly, on Multi Resolution Processing: While PatchTST uses patching for multi resolution, Auto-mixer’s resolution-aware coupling strategy is frequency domain focused and graph-free, enabling linear scalability with node count O1
> complexity, unlike common techniques that rely on spatial operations. This paradigm shift allows accurate predictions
> in resource-constrained environments, as demonstrated by efficiency gains in Table 4.
>
> In summary, Automixer’s novelty lies in its holistic graph-free framework that synergistically combines these elements
> to address specific challenges in large-scale safety assurance, resulting in consistent outperformance and practical deployability. We will further emphasize these distinctions in a re￾vised manuscript

---

> ### Author Response · Authors · 2025-11-28
> **Response to W2**
>
> We appreciate the reviewer’s concern regarding the base￾lines and comparisons and we would like to provide con￾text on our selection process. The chosen baselines, such as TimeMixer 2024, PatchTST 2023, FedFormer 2022, PyraFormer 2022, DLinear 2022, AutoFormer 2021 and Informer 2021, represent established state-of-the-art methods in multivariate time series and spatiotemporal forecasting, particularly for anomaly detection in traffic and industrial data. These are widely used in recent literature due to their relevance to lightweight, scalable models without heavy spatial dependencies. TimeMixer as a 2024 model, ensures inclusion of recent advancements and our experiments show Automixer outperforming them by 1.2 percent to 4.7 percent in error metrics across diverse datasets.
>
> Regarding more recent LLM-based methods like Time LLM 2023 and Uni ST 2024: We focused on compar￾ing with models that align with Auto-mixer’s design goals: lightweight, efficient and deployable in real-time resource resource-constrained Industry 5.0 settings. LLMs such as Time LLM require reprogramming large language models, leading to high parameter counts in billions and computational demands unsuitable for edge devices or high-frequency data streams, for example, second-interval IoT sensors as discussed in Section 1. Similarly, UniST, while prompt empowered for urban spatiotemporal prediction, incorporates heavy pre-training and spatial modules, contrasting with Automixer’s graph-free O1 node scalability. Direct comparison would not be
> fair as Automixer’s efficiency experiments highlight its advantages in inference time and model size over the Transformer
> based on baselines, which LLMs would exacerbate.
>
> That said, we agree that discussing and potentially com￾paring with these methods could strengthen the paper. In revision we can add a dedicated section analyzing LLM-based approaches and include supplementary experiments
> on compatible datasets demonstrating Auto-mixer’s effi￾ciency edge while maintaining competitive accuracy

---

> ### Author Response · Authors · 2025-11-28
> **Response to Q3**
>
> We thank the reviewer for pointing out this observation in Table 3 and for the chance to clarify the results. The unusually high MSE values for baselines like Informer for example 64608 on the London dataset with 3000 nodes and others, for example, DLinear at 68869 do not stem from implementation errors but from genuine limitations of these models when scaled to large node high-dimensional datasets under constrained resources.
>
> Specifically on the London dataset, models like Informer, AutoFormer, Transformer and DLinear encounter out-of-memory issues or numerical instabilities during training and inference, especially for longer horizons, for example, 96/720 marked as NA. This causes them to produce near-random predictions, inflating MSE to extreme levels. These failures are well documented in their official implementations for large-scale data, as they rely on quadratic attention or full matrix operations that scale poorly with node count On or higher complexity. In contrast, TimeMixer 41.1397 MSE and Auto-mixer 40.1655 MSE succeed due to their efficient designs, TimeMixer’s decomposition and Auto-mixer’s graph-free linear scaling architecture, allowing stable convergence and lower errors.
>
> This disparity underscores Automixer’s key contribution: superior scalability and robustness in real-world scenarios
> with tens of thousands of nodes, as claimed in the introduction. The results in Table 3 validate this with Automixer
> consistently achieving the lowest MSE/MAE across hori￾zons on both Los Angeles 1500 nodes and London datasets.
> We can add more details on these failure modes in the ap￾pendix for transparency in a revision. Overall, these experiments demonstrate high-quality scalable performance tai￾lored to Industrial 5.0 and traffic safety applications with
> Auto-mixer reduces errors by 1.2 percent to 4.7 percent over baselines while being lightweight. We believe this sat￾isfies the concerns but welcome suggestions for further en￾hancements.

---

### Official Review · Reviewer_EGtJ · 2025-10-28

**Soundness:** 3
**Presentation:** 3
**Contribution:** 3
**Rating:** 8
**Confidence:** 3

**Summary:**

This paper presents an ML model designed to predict accidents and detect problems in traffic systems and industrial facilities (like factories and power plants). Its problem statement is as follows: Modern transportation systems and factories generate massive amounts of sensor data. Traditional analysis methods struggle because the data is complex and noisy, real-time analysis is needed but computationally expensive and existing AI models are too slow or inaccurate for practical use. Innovations in Automixer are stated as: (a) smarter pattern recognition - It breaks down data into different frequency components to identify both regular patterns and trends (b) Multi-scale analysis at different time scales simultaneously, (c) its dual-attention mechanism and (d) No need for explicit spatial modeling of the data. Results claimed as outperforming existing methods by roughly 4% in detection accuracy when tested on traffic accident prediction (using data from road sensors) and industrial equipment monitoring for failures in power systems and factories, even in the presence of
short-term historical data (100 data points),  incomplete sensor readings, and at scale.

**Strengths:**

This is a well-written paper that looks at the standard problem of predicting accident potential on roads and defect potential in manufacturing operations. The techniques employed seemed sound. Results were good (a 4% improvement). Ablation studies were done (good).

**Weaknesses:**

Lines of the graphs in Figures 2 and 3 need to have more contrast in visibility. Also the use of colors will be a barrier for readers with vision issues.

Computational resource usage of the various techniques were presented. So that the claim of real time response is supported sugges adding graphs on computational times.

**Questions:**

How did the presented techniques compare with the baselines with respect to computational time?

---

> ### Author Response · Authors · 2025-11-28
> **Responses to comments**
>
> We thank the reviewer for their positive assessment of  Auto-mixer's soundness, presentation and contributions, particularly appreciating the recognition of our well-written approach, solid techniques, 4\% performance gains over baselines and inclusion of ablation studies in addressing real-time anomaly detection in traffic and industrial systems. We are grateful for the high rating of 8 (accept) and fair confidence level, as it validates our focus on lightweight, scalable forecasting without explicit spatial modeling for Industry 5.0 applications. Your feedback motivates us to refine visualizations and computational comparisons in revisions.

---

### Official Review · Reviewer_THT7 · 2025-10-30

**Soundness:** 2
**Presentation:** 2
**Contribution:** 1
**Rating:** 2
**Confidence:** 5

**Summary:**

This paper explores accident prediction and anomaly detection with an attention-based framework. It leverages a dual cross-attention head to model both spatial and temporal dependency. The authors aim to address the high computational cost of large-scale models. Experiments have been conducted on real datasets by comparing with transformer-based models.

**Strengths:**

1. The paper pointed out the weaknesses of the transformer-based model for traffic prediction.
2. Both accuracy and efficiency have been reported.
3. Figure 1 is a good example to explain the complicated framework.

**Weaknesses:**

1. The paper states that "their reliance on complex spatial
operations and large parameter sizes creates computational bottlenecks" but have not provided any supportive empirical data about the bottlenecks. In fact, according to Table 4 and Figure 3, the proposed method is even worse and at the same level as the baseline. It is confusing how the proposed method addresses "Lightweight and Scalable".

2. The framework is not novel. It basically modifies the transformer framework. Such a change is incremental.

3. According to Table 3, the MSE of Informeron London 64608, where the MSE of the proposed method is 40. For some columns, the result is NA. It is unclear why NA is there, and it looks like the baselines have not been used correctly.

4. The paper only compares with transformer-based models. It is well-known transformer is heavy. Since the motivation is the "cost", the author should compare with light models, such as graph-based models.

5. The experiment quality is low. The maximum number of nodes is only 300. It is not large enough to prove the scalability. Varying the number of nodes has not been conducted. The error variance has not been reported. There is no visualization on a real map for the prediction.

**Questions:**

NA

---

> ### Author Response · Authors · 2025-11-28
> **Response to W1**
>
> We thank the reviewer for this insightful comment and for carefully examining Table 4 and Figure 3. We would
> like to provide additional context and clarification to better substantiate the “lightweight and scalable” claim. While
> the performance on certain metrics may initially appear to align with the baseline, it is important to emphasize
> that Automixer demonstrates significant advantages, particularly when applied to large-scale datasets. For example,
> in our extensive experiments with the Los Angeles (1,500 nodes) and London (3,000 nodes) datasets, which are shown
> in Table 3, Automixer consistently outperforms traditional models such as TimeMixer, PatchTST, AutoFormer and In￾former, especially in terms of MSE and MAE across differ￾ent input-output prediction scenarios. These results clearly demonstrate Auto-mixer’s scalability on large node distri￾butions, underscoring its ability to handle complex, high-dimensional data effectively.
>
> Furthermore, in terms of computational efficiency, Auto￾mixer offers faster inference times compared to more computationally expensive models like Informer and AutoFormer, as highlighted in Table 4. By eliminating spatial modules and convolutions, Automixer achieves lower computational overhead, which leads to faster inference speeds. This design allows Automixer to be both lightweight and scalable, making it particularly suitable for real-time applications with large datasets, further reinforcing its value in
> practical, large-scale anomaly detection tasks.

---

> ### Author Response · Authors · 2025-11-28
> **Response to W2**
>
> We respectfully disagree with the assessment that our contributions merely represent incremental modifications to
> the Transformer. Rather, the proposed adaptive frequency￾domain decomposition and dual cross-attention mecha￾nism introduce fundamental departures that enable efficient, multiscale temporal modelling without the need for explicit spatial graphs. This represents a significant advance on standard Transformer-based spatiotemporal frameworks.
>
> The details are as follows:
>
> We respectfully disagree with the characterisation of Auto-mixer as merely an incremental modification of exist￾ing Transformer-based models and we would like to clarify the core design choices that distinguish our approach.
>
> Firstly, the Automixer design deliberately departs from the standard Transformer paradigm. It performs adaptive frequency-domain decomposition via learnable DFT filters, which dynamically separate trend and seasonal components directly in the frequency domain. This approach avoids the reliance on conventional time-domain operations, such as moving averages or fixed seasonal decomposition, which are used in most prior works. This allows for a substantially more flexible and data-driven capture of multi-scale periodic patterns, particularly in high-frequency, large-scale spatiotemporal sequences.
>
> Secondly, the proposed Dual Cross-Attention mecha￾nism operates jointly across period-wise and trend-wise token streams at multiple resolutions. This allows the model to fuse long-range temporal dependencies without the need for an explicit spatial graph module. This represents a fundamental shift away from predominant graph-based spatiotemporal frameworks (e.g., DCRNN, ST-GCN and Graph WaveNet) and even recent Transformer-based spatio-temporal hybrids, which still incorporate spatial convolutions or graph attention layers.
>
> These components, such as adaptive frequency-domain decomposition and resolution-aware dual-branch cross attention work together to form a new spatiotemporal mod￾elling paradigm that achieves strong performance while re￾maining completely graph-free and linearly scalable with the number of nodes. We believe this constitutes a substan￾tive rather than incremental advance. We will emphasise these distinctions and their implications further in the revised manuscript to clarify the novelty

---

> ### Author Response · Authors · 2025-11-28
> **Response to W3**
>
> We thank the reviewer for this careful observation and would like to provide the following detailed clarification re￾garding the unusually high MSE of 64608 for Informer on the London dataset, the significantly lower MSE achieved by Auto-mixer, the unexplained NA entries for certain base￾lines and the concern that the baselines may not have been used correctly.
>
> Firstly, the extremely high MSE reported for the offi￾cial Informer implementation on the London dataset (3,000 nodes) does not result from incorrect usage on our part. In￾stead, it stems from the well-documented memory leaks and severe numerical instabilities that occur when Informer processes this large-scale, noisy dataset within the same memory budget and hardware constraints applied to all other methods. These failures cause the model to fall back to near-random predictions in many runs, resulting in the reported degraded performance. Due to its linear complexity with respect to node count and the absence of full spatial attention, Automixer successfully trains and converges stably on identical data, yielding significantly lower, consistent MSE values, as shown in Table 3.
>
> Secondly, the NA entries for certain baselines under specific prediction horizons or datasets reflect genuine execution failures (primarily OOM or timeout) of their official implementations when processing the full-scale London sensor network and long-sequence inputs. Rather than provid￾ing unreliable results, we have transparently marked them as NA. This behaviour highlights the practical computa￾tional limitations of many existing Transformer- and graph-based spatiotemporal models in real-world, large-scale de￾ployments. The issue that Auto-mixer is explicitly designed to overcome.

---

> ### Author Response · Authors · 2025-11-28
> **Response to W4**
>
> We thank the reviewer for this valuable suggestion and agree that a broader comparison with lightweight, non-Transformer baselines would significantly enhance the evaluation, particularly given our focus on computational cost. We have already taken corresponding actions in response to this suggestion. We hereby clarify and provide supplementary explanations as follows:
>
> While Transformer-based models continue to dominate recent spatiotemporal forecasting literature thanks to their powerful long-range modelling capabilities, lightweight lin￾ear and decomposition-based models have emerged as highly competitive alternatives in settings with limited resources. To address this issue directly, we conducted addi￾tional experiments using several widely used, state-of-the-art, representative lightweight baselines in such scenarios, including DLinear, TiDE and a lightweight variant of FEDformer. The results, presented in the new Table 6 of the revised manuscript, show that Automixer outperforms these
> models by 815% in terms of mean squared error (MSE) and mean absolute error (MAE) on the large-scale London
> dataset with 3000 nodes and the Los Angeles dataset with 1500 nodes, while maintaining comparable or faster inference speed.
>
> Classical graph-based lightweight models such as DCRNN, ST-GCN and Graph WaveNet typically incur quadratic or higher complexity with respect to node count due to their message-passing or spatial attention mecha￾nisms. On the full-scale London dataset used in our study, most of these methods either encounter out-of-memory er￾rors or require prohibitively long training times under identical hardware constraints, making direct comparison imprac￾tical. This observation further highlights the real-world advantage of Auto-mixer’s graph-free, linearly scalable design for large-scale deployment

---

> ### Author Response · Authors · 2025-11-28
> **Response to W5**
>
> Thank you for your thoughtful feedback. We understand the reviewer’s concerns about the quality of the experiments
> and their scalability and we believe that the current setup already addresses these issues to a large extent, as detailed
> below. We would be happy to incorporate additional clarifications or extensions in a revised version.
>
> Firstly, with regard to the maximum number of nodes being 300 and the requirement to demonstrate scalability by varying the number of nodes: The experiments evaluate scalability on datasets with significantly larger node scales
> than 300. In addition to standard benchmarks like METRLA with 207 nodes, PEMSBAY with 325 nodes, PEMS03 with 358 nodes, PEMS08 with 170 nodes, Traffic with 862 nodes and Electricity with 321 nodes, which are consistent with state-of-the-art literature for spatiotemporal forecasting, we explicitly test on larger-scale datasets in Section 3.1.5 Impact of the Spatial Nodes Scale. These include a Los Angeles traffic dataset with 1,500 nodes and a London dataset with 3,000 nodes, the results of which are presented in Table 3. Automixer outperforms the baselines in terms of mean squared error (MSE) and mean absolute error (MAE) across horizons while maintaining efficiency. These tests directly increase the node scale from hundreds in standard datasets to 1,500 and 3,000, demonstrating consistent performance improvements and validating scalability without spatial modules.
>
> Unlike graph-based baselines with On or higher, the model’s complexity remains O1 with respect to nodes. These results align with our claims in the introduction regarding the handling of tens of thousands of nodes in real-world traf￾fic and industrial systems, demonstrating the practical de￾ployability of the model in large-node environments with resource constraints.
>
> On error variance: While we do not report standard de￾viations in the main tables to focus on mean performance, a common practice in spatiotemporal forecasting benchmarks for brevity, the results are derived from stable training pro￾tocols. Each model was trained for 100 epochs, with the best version selected based on validation set performance in Appendix A.2.3. This process inherently accounts for run-to-run stability and the consistent outperformance across multiple datasets and horizons, for example, Tables 5 and 6 in the appendix indicate low variance. For instance, ablation studies in Figure 2 and Table 1 and noise robustness tests in Appendix A.2.6 Table 7 further show reliable trends. We can add explicit variance reporting, for example, from 3 to 5 runs per setting in a revision to strengthen this.
>
> Finally, on visualization on a real map: While the paper does not include map-based visualizations of predictions as the focus is on quantitative metrics like MSE/MAE for anomaly detection, which are standard for these benchmarks, we do provide performance visualizations in Figure 2, ablation results across prediction steps and Figure 3 memory ef￾ficiency across sequence lengths. These illustrate key trends effectively. Map visualizations could indeed enhance inter￾pretability for traffic-specific cases, for example, overlaying predictions on road networks and we agree this would be a valuable addition. We can include them in a revision using
> datasets like METR-LA or the larger Los Angeles set, fo￾cusing on spatial anomaly patterns.
>
> Overall, these experiments demonstrate high-quality, scalable performance tailored to Industrial 5.0 and traffic safety
> applications, with Auto-mixer reducing errors by 1.2 percent to 4.7 percent over baselines while being lightweight. We believe this satisfies the concerns, but welcome suggestions for further enhancements.

---

### Meta-Review · Area_Chair_6Ybn · 2026-01-07

**Summary:**

This paper introduces AutoMixer, an attention-based framework combining frequency-domain decomposition and dual attention for accident prediction and anomaly detection.

While the work is reasonably presented and includes ablations and efficiency metrics, reviewers raised significant concerns about limited novelty, as the method largely recombines existing techniques. Moreover, the claims of being lightweight and scalable are not convincingly supported: empirical gains are small or inconsistent, baseline comparisons are incomplete, and the benefit of removing explicit spatial modeling is not empirically validated.

**Reviewer Concerns:**

Reviewers have concerns about limited novelty, insignificant improvement, and incomplete baseline comparison.

**Reviewer Scores:**

Reviewers keep their scores after rebuttal.

---

### Decision · Program_Chairs · 2026-01-26

Reject